# Enhancing Hallucination Detection through Noise Injection

## Abstract

Large Language Models (LLMs) are observed to generate plausible yet incorrect responses, known as hallucinations. Effectively detecting such hallucination instances is crucial for the safe deployment of LLMs. Recent research has linked hallucination to model uncertainty, suggesting that hallucinations can be detected by measuring dispersion over answer distributions obtained from a set of samples drawn from the model. While using the model's next token probabilities used during training is a natural way to obtain samples, in this work, we argue that for the purpose of hallucination detection, it is overly restrictive and hence sub-optimal. Motivated by this viewpoint, we perform an extensive empirical analysis showing that an alternative way to measure uncertainty - by perturbing hidden unit activations in intermediate layers of the model - is complementary to sampling, and can significantly improve detection accuracy over mere sampling.

## 1 Introduction

Large Language Models (LLMs) have made significant advancements in recent years (Achiam et al., 2023; Zhao et al., 2023). However, despite the strides, LLMs are observed to sometimes generate plausible yet incorrect responses – a phenomenon known as hallucination (Ji et al., 2023; Kuhn et al., 2023a). To ensure the safe deployment of LLMs, effective detection of hallucination is essential, and it has gained significant research attention (Malinin & Gales, 2020; Lin et al., 2022; 2023; Kuhn et al., 2023a; Chen et al., 2024). Many research efforts focus on detecting hallucinations by assessing model uncertainty across samples drawn from the model. For example, Malinin & Gales (2020) proposes leveraging predictive uncertainty for hallucination detection. Similarly, Lin et al. (2022) and Lin et al. (2023) propose semantic consistency and quantify lexical similarity across samples. The core principle underlying this line of work is simple: the greater the observed uncertainty, the higher the likelihood of hallucination.

Since a language model defines the probability distribution over the next tokens, the most obvious way to generate such samples is therefore to repeatedly sample from the conditional distribution over tokens given the context so far. A benefit of this way of sampling is that it stays faithful to the probability distribution defined by the model (up to any deviations from the training temperature). Generating faithful samples from the model furthermore makes sense, in particular, when the goal is to generate individual answers, say, to a given prompt.

We note, however, that in the case of hallucination detection, the purpose of sampling is not to generate standalone answers, but to estimate the coherence of a model's responses to a given prompt. The above-mentioned approaches can in this context also be viewed as

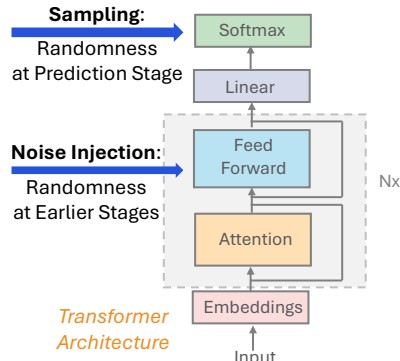

Figure 1: **Source of Randomness in Hallucination Detection**. Prior work uses prediction layer sampling and measures model uncertainty across samples for hallucination detection. Additionally, we explore noise injection that randomly perturbs intermediate representations, introducing a second source of randomness at earlier stages.

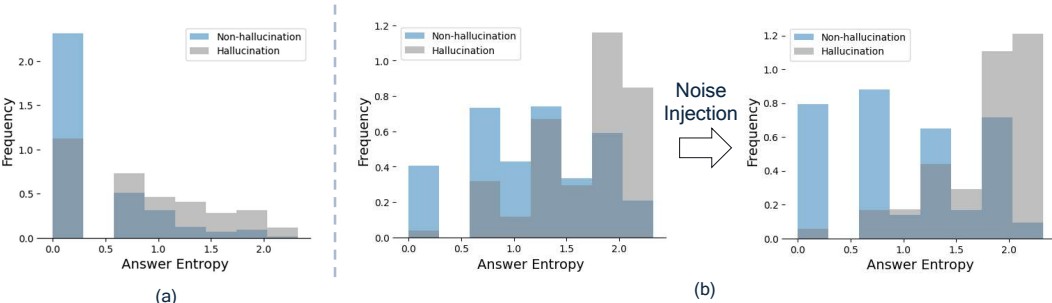

Figure 2: **Effect of Intermediate Layer Randomness on Hallucination Detection.** *(a) Standalone Effect.* With noise injected to randomly perturb intermediate representations, LLM exhibits greater uncertainty when hallucination (grey) compared to non-hallucination (blue); *(b) Combined Effect.* Injecting noise improves hallucination/non-hallucination separation, enhancing hallucination detection effectiveness. *(b) Left:* prediction layer sampling alone; *(b) Right:* noise injection and prediction layer sampling. Model uncertainty measured by Equation 4. A higher value indicates a higher uncertainty level. Evaluation performed on GSM8K dataset with `Llama2-13B-chat` model across 5 generations.

performing a type of sensitivity analysis that makes it possible to assess the likelihood of a given prompt to elicit a hallucination in a model. A distribution of responses that stays coherent under perturbations is considered as evidence for the model to "know" the correct response for a given prompt, and for an answer generated by the model accordingly to be truthful.

It is commonly assumed in language modeling that hidden unit activations tend to capture the more abstract and high-level representations of a given phrase or thought, while logits and low-level token embeddings capture representations that reduce it to a specific syntactic form. This suggests that, even though it is tempting to rely on sampling from the model to assess coherence for a given prompt, a better way to assess coherence should involve perturbations of these hidden representations. Unlike sampling, which preserves the token likelihood order regardless of the sampling temperature, hidden representation perturbation can disrupt this order by altering token probabilities. These distinct impacts suggest that perturbing hidden representations could provide a complementary view of coherence, particularly for hallucination detection.

To this end, we study model behavior under randomness introduced in earlier stages of LLM computation. Particularly, we inject noise to perturb intermediate layer representations, as illustrated in Figure 1. Under noise perturbation, we hypothesize that a model would exhibit higher uncertainty when hallucinating, consistent with the relationship between model uncertainty and hallucination found in prior research. We empirically validate the hypothesis in Figure 2 (a), where hallucination cases (grey) show higher variance under noise injection, reflected by higher entropy. Additionally, we examine the interplay between intermediate layer noise injection and the prediction layer sampling. Since two sources of randomness operate at different layers, we hypothesize and validate that they have complementary effects on the model uncertainty, as shown in Figure 3. Based on our observation, we propose combining intermediate layer noise injection with prediction layer sampling to enhance hallucination detection. We empirically validate that this combination improves the separation between hallucination and non-hallucination instances in terms of model uncertainty in Figure 2 (b). Extensive experiments demonstrate the effectiveness of noise injection in enhancing hallucination detection across various datasets, uncertainty metrics, and model architectures such as `Llama2-7B-chat`, `Llama2-13B-chat`, and `Mistral`.

## 2 PROBLEM STATEMENT

Prior work (Malinin & Gales, 2020; Lin et al., 2022; 2023; Kuhn et al., 2023a; Chen et al., 2024) connects hallucination detection to model uncertainty estimation. Given an uncertainty metric $E(\cdot)$, detecting whether the model is hallucinating for a given input context $x$ can be framed as a binary classification problem:

$$D(x) = \begin{cases} \text{Non-Hallucination} & \text{if } E(\mathcal{Y}) < \tau \\ \text{Hallucination} & \text{if } E(\mathcal{Y}) \geq \tau \end{cases},$$

where $\tau$ is the threshold and $\mathcal{Y} = \{\boldsymbol{y}^1, \boldsymbol{y}^2, \ldots, \boldsymbol{y}^K\}$ denotes $K$ generations for the given input context. A higher level of uncertainty indicates model hallucination.

**Uncertainty Metric**  One critical aspect of hallucination detection is the design of uncertainty metrics $E(\cdot)$ over generations $\mathcal{Y}$. A commonly used metric is *Entropy*, computed from the sequence joint distribution:

$$E_{raw}(\mathcal{Y}) = -\mathbb{E}_{\boldsymbol{y} \in \mathcal{Y}} \sum_{t=1}^{T} \log p(y_t \mid y_{<t}, \boldsymbol{x}) \tag{1}$$

However, entropy can be biased against longer sequences due to smaller joint probabilities. To address this, Malinin & Gales (2020) proposes *Length Normalized Entropy*:

$$E_{normalized}(\mathcal{Y}) = -\mathbb{E}_{\boldsymbol{y} \in \mathcal{Y}} \frac{1}{T_{\boldsymbol{y}}} \sum_{t=1}^{T} \log p(y_t \mid y_{<t}, \boldsymbol{x}) \tag{2}$$

For reasoning tasks, we also consider an uncertainty metric focused on the answer space, as detailed in Section 3.1. The metric targets the final answer rather than intermediate tokens, making it particularly well-suited for reasoning tasks with lengthy intermediate steps.

**Source of Randomness**  To effectively quantify model uncertainty requires not only an uncertainty metric $E(\cdot)$ but also a sufficiently diverse set of generations $\mathcal{Y}$, necessitating the introduction of randomness during generation. Prior work typically introduces randomness only at the final prediction stage by sampling from the next token distribution $p(y_t \mid y_{<t}, \boldsymbol{x})$. In addition, we introduce randomness at earlier stages.

Consider a typical LLM consisting of an embedding layer, a stack of $L$ transformer layers, and a prediction layer $W$. At each decoding step $t$, intermediate representations $\boldsymbol{h}_t^l$ are computed layer by layer for a given input $\boldsymbol{x}$. The next token probability $p(y_t \mid y_{<t}, \boldsymbol{x})$ explicitly conditioned on $\boldsymbol{h}_t^L$ (and $\boldsymbol{h}_t^{L-1}$ via skip connections) but is implicitly affected by earlier layers, as they shape these final representations. This relationship can be expressed as:

$$p(y_t \mid y_{<t}, \boldsymbol{x}) = f(\boldsymbol{h}_t^1, \ldots, \boldsymbol{h}_t^L). \tag{3}$$

We inject noise to perturb the intermediate representation at layers $l_1$ through $l_2$. As a result, given noise $\epsilon$, the next token distribution is stochastically modified as

$$\tilde{p}(y_t \mid y_{<t}, \boldsymbol{x}, \epsilon) = f(\boldsymbol{h}_t^1, \ldots, \tilde{\boldsymbol{h}}_t^{l_1}, \ldots, \tilde{\boldsymbol{h}}_t^{l_2}, \ldots, \boldsymbol{h}_t^L),$$

where each $\tilde{\boldsymbol{h}}_t^l$ is a noise-perturbed version of $\boldsymbol{h}_t^l$. Notably, for $l' > l_1$, $\boldsymbol{h}_t^l$ is computed from the perturbed representations of prior layers. With noise sampled from $g(\epsilon)$ and randomized across generations, sampling from $\tilde{p}(y_t \mid y_{<t}, \boldsymbol{x}, \epsilon)$ at each generation combines randomness at the prediction and intermediate layer.

## 3 INTERMEDIATE LAYER RANDOMNESS AND HALLUCINATION DETECTION

In this section, we conduct a case study to investigate LLM behavior under intermediate layer randomness. We first hypothesize and validate that, with noise injected to modify intermediate layer representations, model responses exhibit greater variability when the model hallucinates. We then observe that intermediate layer noise injection has a complementary effect on model uncertainty compared to prediction layer sampling. Based on our observations, we propose to combine noise injection with prediction layer sampling to enhance hallucination detection.

### 3.1 CASE STUDY SETUP

We focus this case study on mathematical reasoning tasks using the GSM8K (Cobbe et al., 2021) dataset. We experiment with the GSM8K test set, containing 1319 questions, using in-context learning examples from Wei et al. (2022). As shown in Table 1, following in-context learning examples, LLM can produce coherent yet incorrect answers—i.e., hallucinations—highlighting the need for

Table 1: **Example of Answer Entropy Computation on GSM8K dataset.** For each response, the answer string is marked in **bold**, with the remaining text representing the reasoning part. We estimate uncertainty by counting the occurrence of each answer string. In this example, with $K = 3$ responses, $E_{answer}(\mathcal{Y}) = -0.67 \times \log 0.67 - 0.33 \times \log 0.33$.

| Responses for question: "*A robe takes 2 bolts of blue fiber and half that much white fiber. How many bolts in total does it take?*" | Answer | Answer Frequency |
|---|---|---|
| Half of 2 bolts of white fiber is 2/2 = 1 bolt. So, it takes 2 + 1 = 3 bolts in total. The answer is **3**. | 3 | 67% |
| 2 bolts of blue fiber and half that much white fiber is 2 + half of 2 = 2 + 1 = 3 bolts. The answer is **3**. | 3 | |
| 2 bolts of blue fiber and half that much white fiber is 2 x 2 = 4 bolts of blue fiber. The answer is **4**. | 4 | 33% |

effective hallucination detection in such reasoning tasks. This extends beyond prior work on hallucination detection (Malinin & Gales, 2020; Lin et al., 2022; 2023; Kuhn et al., 2023a; Chen et al., 2024), which primarily focuses on question-and-answer tasks such as TriviaQA (Joshi et al., 2017). Section 4 demonstrates that our algorithm also generalizes to knowledge-based question-and-answer tasks.

GSM8K consists of mathematical question-response pairs $\{\boldsymbol{x}, \boldsymbol{y}\}$, where each response includes both the reasoning and the answer: $\boldsymbol{y} = [\boldsymbol{r}, \boldsymbol{a}]$. As shown in Table 1, the reasoning chains for GSM8K can be lengthy, yet the final answer is more critical. Therefore, treating all tokens equally in uncertainty estimation, as in Equations 1 and 2, can be less effective. To address this, we estimate uncertainty by counting the occurrences of each answer string and introduce the metric of *Answer Entropy*:

$$E_{answer}(\mathcal{Y}) = -\sum_j p(\boldsymbol{a}_j) \log p(\boldsymbol{a}_j) \tag{4}$$

where $p(\boldsymbol{a}_j)$ is the empirical probability of each unique answer $\boldsymbol{a}_j$ over the $K$ final answers $\{\boldsymbol{a}^1, \boldsymbol{a}^2, \ldots, \boldsymbol{a}^K\}$ extracted from $K$ responses $\mathcal{Y} = \{\boldsymbol{y}^1, \boldsymbol{y}^2, \ldots, \boldsymbol{y}^K\}$. An example of answer entropy computation is provided in Table 1.

Our case study focuses on the `Llama2-13B-chat` model, where uniform noise sampled from $U(0, 0.05)$ to additively perturb the MLP layer outputs of $25 - 40$ transformer layers. We follow the default generation configuration with top-k = 50 and top-p = 1. When prediction layer sampling is enabled, we set temperature as $T = 0.8$, which optimizes GSM8K accuracy within the set $T = \{0.2, 0.5, 0.8, 1.0\}$. Experiments involving alternative datasets, uncertainty functions, models, injection layers, and noise types are discussed in Section 4.

### 3.2 HALLUCINATION INCREASES RESPONSE VARIABILITY UNDER NOISE INJECTION

In this study, we investigate how LLMs behave under noise injection in intermediate layers as the sole source of randomness. Given that prior research indicates model uncertainty increases during hallucination, we hypothesize that the model's response will exhibit greater variability when hallucinating. To validate our hypothesis, at each decoding step, we perturbed the MLP output of $25 - 40$ transformer layers as $\tilde{\boldsymbol{h}}_t^l = \boldsymbol{h}_t^l + \epsilon$, with $\epsilon$ is uniformly sampled from $U(0, 0.05)$. The next token prediction is thus stochastically modified at each generation as $\tilde{p}(y_t \mid y_{<t}, \boldsymbol{x}, \epsilon) = f(\boldsymbol{h}_t^1, \ldots, \boldsymbol{h}_t^{24}, \tilde{\boldsymbol{h}}_t^{25}, \ldots, \tilde{\boldsymbol{h}}_t^{40})$. To isolate the effect of noise injection, we set the sampling temperature to zero and greedily select the next token with the largest likelihood, removing randomness from the prediction layer sampling process.

To assess model uncertainty under the noise injection, we generate $K = 5$ responses for each question and compute answer entropy following 4. We classify model hallucination on a question level and model responses to a question are considered as hallucinating if the majority of the $K = 5$ generated answers are incorrect, and as non-hallucinating otherwise. In Figure 2 *Left*, we compare answer entropy between hallucinating and non-hallucinating cases by overlaying the histograms of the two groups. We observe that the model exhibits greater variability under noise when halluci-

nating (grey), as evidenced by higher entropy values. This observation matches our intuition: less variability implies the robustness of the model response to noise, suggesting greater certainty and a lower likelihood of hallucination.

## 3.3 COMPLEMENTARY EFFECT OF NOISE INJECTION AND PREDICTION LAYER SAMPLING

We now extend our investigation beyond a single source of randomness. Particularly, we study the interplay between noise injection and the standard source of randomness – prediction layer sampling. Since the two sources of randomness operate at different layers with distinctive roles in model prediction, we hypothesize that they would have complementary effects on model uncertainty.

This hypothesis is theoretically grounded in the distinct impacts of each randomness source: prediction layer sampling preserves token likelihood ordering for any temperature. In contrast, noise injection perturbs intermediate representations, potentially reversing token orderings. These distinct mechanisms operate at different stages, suggesting complementary effects on model uncertainty.

To test our hypothesis, we compare model uncertainty under two sources of randomness.

**Intermediate Layer Noise Injection:** We follow the setup outlined in Section 3.2, injecting noise sampled from $U(0, 0.05)$ and setting the temperature to zero.

**Prediction Layer Sampling:** We do not perturb model computation; instead we sample with temperature $T = 0.8$ from the unmodified next token probability $p(y_t \mid y_{<t}, \boldsymbol{x}) = f(\boldsymbol{h}_t^1, \ldots, \boldsymbol{h}_t^{40})$. The non-zero temperature introduces sampling randomness at the prediction layer, with $T = 0.8$ selected to maximize model accuracy.

For each setup, we assess model uncertainty across $K = 50$ generations for each question following Equation 4. We then compare the model uncertainty under two sources of randomness, as illustrated in Figure 3. The scatter plot displays each question of the GSM8K test set as a point, with the x-value representing model uncertainty under prediction layer sampling alone, whereas the y-value represents model uncertainty under intermediate layer noise injection. The plot reveals that model uncertainty under the two sources of randomness is related but not identical, with a Pearson correlation (Sedgwick, 2012) of 0.67. This indicates a positive correlation but also highlights the complementary effects between the two randomness sources. We further validate the complementary effect in Section 4.3

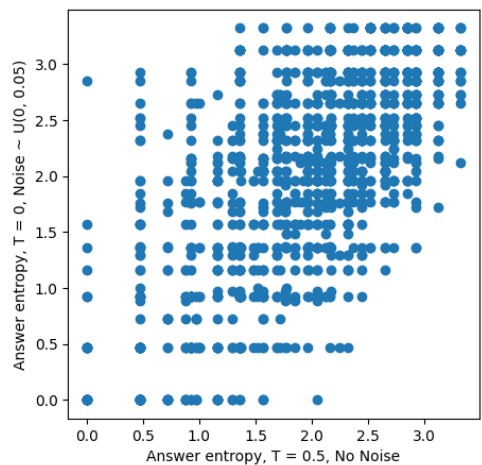

Figure 3: **Complementary Effect of Different Randomness Sources**. The x-axis presents model uncertainty with prediction layer sampling whereas the y-axis presents model uncertainty under intermediate layer noise injection. A Pearson correlation of 0.67 indicates a complementary relationship between the two sources.

## 3.4 ALGORITHM: NOISE INJECTION AS A HALLUCINATION DETECTION AMPLIFIER

To leverage the complementary effect of different sources of randomness revealed in Section 3.3, we incorporate noise injection alongside prediction layer sampling and propose our Noise Enhanced Hallucination Detector. The design is illustrated with additive uniform noise in Algorithm 1.

Specifically, for a given noise magnitude $\alpha$ and a set of layers $l^1$ through $l^2$, we inject additive uniform noise $\boldsymbol{\epsilon} \sim U(0, \alpha)^d$ to the MLP output of the selected layers, where $d$ is the model dimension. At each decoding step, the selected layers are perturbed as $\tilde{\boldsymbol{h}}_t^l = \boldsymbol{h}_t^l + \epsilon$, where $\boldsymbol{h}_t^l$ with $l' > l^1$ is computed from the perturbed representations of prior layers. This perturbation stochastically modifies the next token probability as $\tilde{p}(y_t \mid y_{<t}, \boldsymbol{x}, \epsilon) = f(\boldsymbol{h}_t^1, \ldots, \tilde{\boldsymbol{h}}_t^{l_1}, \ldots, \tilde{\boldsymbol{h}}_t^{l_2}, \ldots, \boldsymbol{h}_t^L)$. Across

Table 2: **Case Study: Effectiveness of Noise Injection for Enhancing Hallucination Detection.** Noise injection (first row) improves detection effectiveness compared to no noise (second row), as indicated by a higher AUROC, without degrading model accuracy. Evaluation on GSM8K dataset with `Llama2-13B-chat` model across 5 generations.

|  | AUROC | ACC |
|---|---|---|
| Answer Entropy w/ T = 0.8, no noise | 73.86 | 34.95 |
| Answer Entropy w/ T = 0.8, noise $\sim$ U(0, 0.05) | 79.12 | 36.32 |

generations, we sample noise $\epsilon$ independently and draw samples from the temperature-adjusted distribution $\tilde{p}_T(y_t \mid y_{<t}, \boldsymbol{x}, \epsilon)$ with temperature $T$. Effectively, our sampling process integrates over noise and follows the marginal distribution

$$\tilde{p}(y_t \mid y_{<t}, \boldsymbol{x}) = \int_{\epsilon} \tilde{p}_T(y_t \mid y_{<t}, \boldsymbol{x}, \epsilon) g(\epsilon),$$

where $g(\epsilon)$ is the probability density function of $U(0, \alpha)^d$. By perturbing the intermediate layer outputs and sampling with a non-zero temperature at the final layer, our approach effectively combines two complementary sources of randomness. To identify hallucinations, we compute the hallucination detection score over $K$ generations and apply a threshold to classify outputs.

---

**Algorithm 1** Noise Enhanced Hallucination Detection

---

**Input:** Input context: $\boldsymbol{x}$, noise magnitude $\alpha$, number of generations $K$, sampling temperature $T$, perturbed layers $l_1$ to $l_2$, uncertainty metric $E(\cdot)$.
**Output:** Hallucination detection score: $s(\boldsymbol{x})$
1: **for** each generation $k = 1$ to $K$ **do**
2:     Sample noise $\boldsymbol{\epsilon} \sim U(0, \alpha)^d$
3:     **for** each decoding step $t$ **do**
4:         **for** each layer l **do**
5:             Compute $\boldsymbol{h}^l$ using the potentially perturbed prior layer representations.
6:             Perturb the MLP outputs: $\tilde{\boldsymbol{h}}^l = \boldsymbol{h}^l + \boldsymbol{\epsilon}$ if $l \in [l_1, l_2]$.
7:         **end for**
8:         Modify next token probability:

$$\tilde{p}(y_t \mid y_{<t}, \boldsymbol{x}, \epsilon) = f(\boldsymbol{h}_t^1, \ldots, \tilde{\boldsymbol{h}}_t^{l_1}, \ldots, \tilde{\boldsymbol{h}}_t^{l_2}, \ldots, \boldsymbol{h}_t^L)$$

9:         Sample token $y_t$ from $\tilde{p}(y_t \mid y_{<t}, \boldsymbol{x}, \epsilon)$ with temperature T, append it to generation $\boldsymbol{y}^k$.
10:     **end for**
11: **end for**
12: **return** Hallucination detection score $s(\boldsymbol{x}) = E(\mathcal{Y})$, where $\mathcal{Y} = \{\boldsymbol{y}^1, \boldsymbol{y}^2, \ldots, \boldsymbol{y}^K\}$

---

In Table 2, we validate the effectiveness of our scheme under the case study setup. We perturb the MLP outputs of layers 25 to 40 with additive uniform noise of magnitude $\alpha = 0.05$, sampled from $U(0, 0.05)$, and evaluate over $K = 5$ generations. In practice, the noise magnitude can be selected based on the validation set, and we present an ablation study on different noise magnitudes in Section 4.3. Following established literature (Malinin & Gales, 2020; Lin et al., 2022; 2023; Kuhn et al., 2023a; Chen et al., 2024)., we assess the effectiveness of hallucination detection using the threshold-free metric, the area under the receiver operating characteristic curve (AUROC), where a higher value indicates better detection performance. As shown in Table 2, our scheme effectively detects hallucination instances with AUROC value $> 50$.

We further compare our scheme with prior schemes which solely rely on prediction layer sampling without noise injection during model computation. The setup of the noiseless scheme follows Section 3.3. As shown in Table 2, our scheme with noise injection significantly improves detection effectiveness and achieves a higher AUROC value. Additionally, this performance enhancement is visualized in Figure 2 *(b)*, where noise injection increases the separation and reduces the overlap in the histograms from left to right.

Table 3: **Intermediate Layers Noise Injection Enhances Hallucination Detection across Diverse Datasets and Uncertainty Metrics.** Hallucination detection AUROC reported, the higher the better. Noise magnitude fixed as $\alpha = 0.05$ based on GSM8K performance. Evaluation with `Llama2-13B-chat` model across 5 generations.

|  | GSM8K | CSQA | TriviaQA | ProntoQA |
|---|---|---|---|---|
| Predictive Entropy | 62.79 | 57.88 | 75.28 | 63.28 |
| Predictive Entropy w/ noise | 62.48 (-0.31) | 58.16 (+ 0.28) | 75.48 (+ 0.20) | 64.36 (+ 1.08) |
| Normalized Entropy | 62.36 | 56.57 | 75.66 | 62.97 |
| Normalized Entropy w/ noise | 62.36 | 56.96 (+ 0.39) | 75.99 (+ 0.33) | 63.95 (+ 0.98) |
| Answer Entropy | 73.15 | 68.11 | 62.82 | 65.07 |
| Answer Entropy w/ noise | 78.55 (+ 5.40) | 69.87 (+ 1.76) | 64.08 (+ 1.26) | 66.68 (+1.59) |

Further, we evaluate model accuracy on the GSM8K dataset based on majority vote, both with and without noise injection. As shown in Table 2, noise injection can boost model accuracy. This supports our intuition that incorrect answers produced during hallucination are less robust to noise injection , as indicated by higher entropy. Consequently, the consistency of incorrect answers across generations reduces with noise injected, making them less likely to be selected by majority vote. This shift improves the likelihood of correct answers being chosen, thereby enhancing accuracy under the majority vote scheme.

## 4 EXPERIMENTS

In this section, we move beyond the case study and extensively validate the effectiveness of our algorithm across different datasets, uncertainty metrics, and model architectures. Further, we conduct a comprehensive ablation study to understand the effect of the number of generations, injection layers, sampling temperature, and noise magnitude.

### 4.1 GENERALIZABILITY ACROSS DIVERSE DATASETS AND UNCERTAINTY METRICS

In addition to mathematical reasoning tasks, we validate our hypothesis on question-and-answer datasets including TriviaQA (Joshi et al., 2017), CSQA (Talmor et al., 2019), and ProntoQA (Saparov & He, 2023). For TriviaQA, we utilize the validation portion of the `rc.nocontext` subset, which contains $9,960$ unique questions. The `rc.nocontext` subset of TriviaQA is designed for question-answering tasks without providing additional context from the source documents. For CSQA, we use the validation set containing $1,221$ questions related to commonsense world knowledge in a multiple-choice format. Following the methodology of Wei et al. (2022), we include their hand-written 7-shot chain-of-thought exemplars for evaluation. PrOntoQA is a synthetic question-answering dataset comprised of procedurally-generated symbolic world models and reasoning chains to resolve the truthfulness of a claim. We extract the generated questions and ground truth reasoning chains for the 1-Hop fictional subset from their provided model outputs, totaling $400$ question-answer pairs.

For each dataset, we select the temperature within $T = \{0.2, 0.5, 0.8, 1.0\}$ which optimizes the model accuracy on this dataset. For GSM8K, TriviaQA, CSQA, and ProntoQA, the temperature is set to be $0.8$, $0.2$, $0.8$, and $0.8$, respectively. We follow the setup of Section 3.1 and select the noise magnitude as $\alpha = 0.05$ based on GSM8K performance. We remark that $\alpha = 0.05$ is not the optimal noise magnitude for each dataset and performance can be further boosted through hyper-parameter search, as demonstrated in Appendix A. For each dataset, we evaluate with uncertainty metrics: Predictive Entropy (see Equation 1), Normalized Predictive Entropy (see Equation 2), and Answer Entropy (see Equation 4). Looking into Table 3, noise injection is most effective on GSM8K with answer entropy, as expected since it is the optimized metric. However, our method remains effective across most datasets and metrics, validating that noise injection generally enhances model performance across various uncertainty metrics.

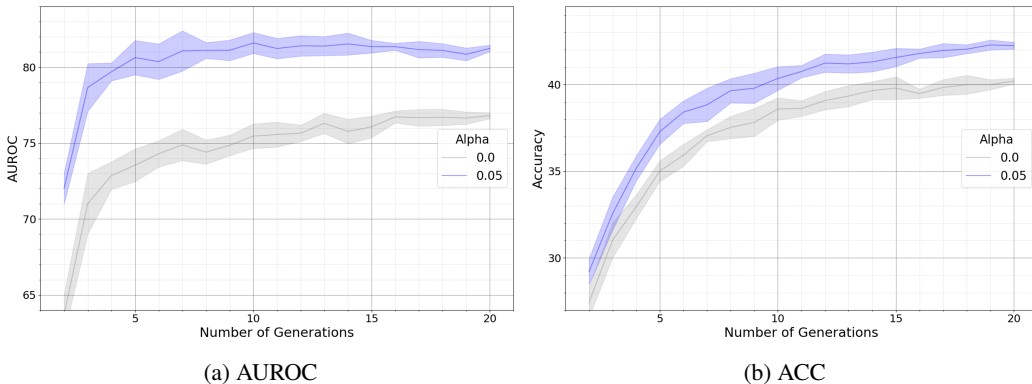

(a) AUROC                                    (b) ACC

Figure 4: **Noise Injection Enhances Hallucination Detection without Degrading Model Accuracy Across Different Number of Generations.** Evaluation with GSM8K datasets on `Llama2-13B-chat` model across 1 - 20 generations. Hallucination detection AUROC (a) and model accuracy (b) reported; higher values are better. The mean and standard deviation across random seeds are shown in the plot.

Table 4: **Ablation on Temperature and Noise Magnitude.** Noise injection (right two columns) improves detection effectiveness compared to no noise (left column), as indicated by a higher AUROC. Evaluation on GSM8K dataset with `Llama2-13B-chat` model across 5 generations.

|  | noise magnitude $= 0$ | noise magnitude $= 0.01$ | noise magnitude $= 0.05$ |
|---|---|---|---|
| T = 0.2 | 71.01 | 74.97 | 75.22 |
| T = 0.5 | 75.98 | 79.59 | 79.38 |
| T = 0.8 | 73.70 | 79.39 | 80.72 |
| T = 1.0 | 66.65 | 79.90 | 76.68 |

## 4.2 ABLATION ON NUMBER OF GENERATIONS

So far, we have presented results based on $K = 5$ generations in Section 3 and Section 4.1. We now extend this study to explore the effect of noise injection across different numbers of generations. In Figure 4, we present the hallucination detection AUROC (left) and model accuracy on GSM8K (right) for $K = 1$ to $K = 20$ generations. The rest of the setup follows Section 3.1. For each $K$, we report the mean and standard deviation across 20 groups of $K$ runs. As shown in Figure 4, both hallucination detection AUROC and model accuracy on GSM8K improve with an increasing number of generations. Notably, noise injection consistently enhances the effectiveness of hallucination detection across different numbers of generations without degrading model accuracy. In practice, the number of generations can be adjusted based on the computational budget and accuracy requirements. Nevertheless, our experiments demonstrate that noise injection improves hallucination detection effectiveness, regardless of the specific number of generations used.

## 4.3 ABLATION ON SAMPLING TEMPERATURE AND NOISE MAGNITUDE

In Section 4.1, we select the temperature temperature per dataset based on model accuracy and set the noise magnitude to 0.05. Table 4, further explores the effect of varying sampling temperature and noise magnitude. The rest of the experiment setup follows Section 3.1. As shown in Table 4, while the optimal noise magnitude varies with temperature, moderate noise injection generally enhances hallucination detection. Additionally, the table highlights the complementary effects of noise and temperature. As randomness increases from $T = 0.8$ to $T = 1.0$ without noise, hallucination detection AUROC drops. Yet injecting noise at $T = 0.8$, adds a different source of randomness and improves performance.

## 4.4 ABLATION ON NOISE INJECTION LAYERS

We now investigate the effect of noise injection on different layers across the LLAMA-13B architecture, which has 40 layers in total. In addition to the upper layers noise (25 - 40 layers) injection,

Table 5: **Noise injection across all layers enhances performance**, with the upper layer demonstrating the greatest effectiveness. AUROC and ACC reported. The higher the values, the better. Evaluation on GSM8K dataset with `Llama2-13B-chat` model across 5 generations.

| | No Noise | Lower Layer Noise | Middle Layer Noise | Upper Layer Noise |
|---|---|---|---|---|
| AUROC | 73.15 | 78.70 | 79.36 | 78.55 |
| ACC | 35.07 | 35.48 | 36.00 | 36.65 |

Table 6: Noise injection improves hallucination detection on `Llama2-7B-chat` and `Mistral`. Evaluation of GSM8K across 5 generations. AUROC value reported; the higher the better.

| | Llama2-7B-chat | Mistral |
|---|---|---|
| No Noise | 75.09 | 77.03 |
| Noise Injection | 76.80 | 82.95 |

we studied so far, we experiment with middle layers (15 - 25 layers) and lower layers (0 - 15 layers) noise injection. In Table 5, we report the hallucination detection AUROC with noise injected on different layers. The noise magnitude is set to 0.05, 0.02, 0.01 for upper layers, middle layers, and lower layers, respectively, each achieving the optimal performance across noise injection level $\{0.01, 0.02, 0.03, 0.04, 0.05\}$ for the corresponding layers. As we observe from Table 5, while noise injection enhances hallucination across layers, upper-layer injection is the most effective. This may be because upper layers tolerate more noise without disrupting generation, reflected by the higher optimal noise magnitude. In contrast, lower layers have less tolerance due to error propagation.

### 4.5 ABLATION ON ALTERNATIVE ARCHITECTURES

We extend our case study beyond the `Llama2-13B-chat` model, experimenting with the `Llama2-7B-chat` from the same Llama family and the `Mistral-7B` model (Jiang et al., 2023) from a different family. Both models have 32 layers in total, and we inject noise into layers 22 to 32 to perturb the upper layer representations. We evaluate GSM8K, following the setup from our case study in Section 3.1. As shown in Table 6, on both architectures, noise injection improves the AUROC of hallucination detection. Notably, the effective noise magnitude differs: while `Llama2-7B-chat` performs well with $\alpha = 0.05$, `Mistral-7B` requires a smaller noise level of $\alpha = 0.02$, indicating the need for model-specific hyperparameter tuning.

### 4.6 ALTERNATIVE UNCERTAINTY METRIC

In addition to the uncertainty metrics defined in Section 2, we investigate other metrics including Lexical Similarity (Lin et al., 2022; 2023) and Semantic Entropy Kuhn et al. (2023b). Lexical Similarity is an uncertainty metric used to gauge how similar text samples are. It specifically calculates the average Rouge-L score across a set of sampled answers $\mathcal{Y} = \{\boldsymbol{y}^1, \boldsymbol{y}^2, \ldots, \boldsymbol{y}^K\}$ for a given context $\boldsymbol{x}$ as $\frac{1}{C} \sum_{i=1}^{K} \sum_{j=i+1}^{K} RougeL(\boldsymbol{y}^i, \boldsymbol{y}^j)$ where $C = K * (K - 1)/2$. Semantic entropy combines the uncertainties of individual tokens within groups of similar meanings. To calculate it, first, the generated outputs are grouped into clusters that share the same semantic meaning. Then, the semantic entropy is determined by summing up the uncertainties within each cluster.

Among the datasets analyzed, only TriviaQA is appropriately suited for evaluating Lexical Similarity and Semantic Entropy. The True/False format of ProntoQA and the multiple-choice format of CSQA are not conducive to Rouge-L measurement. Similarly, the numerical answers in GSM8K are incompatible with the clustering required for Semantic Entropy analysis. Conversely, the short, free-form answers in TriviaQA make it an ideal candidate for both metrics.

In Table 7, we present the AUROC numbers for Lexical Similarity and Semantic Entropy on TriviaQA, evaluated at a temperature of 0.2 and noise magnitudes of $\alpha = 0$ and $\alpha = 0.05$. The data clearly indicate that both uncertainty metrics show improvement following the introduction of noise.

Table 7: Noise Injection Enhances Hallucination Detection under Lexical Similarity and Semantic Entropy. Evaluation on TriviaQA dataset with `Llama2-13B-chat` model across 5 generations.

|  | Lexical Similarity | Semantic Entropy |
|---|---|---|
| Noise = 0 | 64.74 | 63.62 |
| Noise ~ U (0,0.05) | **66.59** | **65.51** |

## 5 RELATED WORK

Several recent works have demonstrated a strong correlation between model uncertainty and the likelihood of hallucination. Measures of model uncertainty include the entropy of answer (Malinin & Gales, 2021), semantic (Kuhn et al., 2023a; Chen et al., 2024; Farquhar et al., 2024), predictive (Xiao & Wang, 2021), and lexical (Lin et al., 2022; 2023) distributions. These methods rely on a diverse set of model generations which primarily used temperature-based sampling techniques. Our work is complementary to these approaches and introduces an additional source of randomness.

In addition to entropy-based estimates, intermediate model activations have been shown to provide insights into model confidence. Chuang et al. (2023) demonstrates that the divergence in activations between correct and incorrect tokens tends to increase across layers, with contrasted activations growing sharper for correct tokens. Additionally, Li et al. (2024) shows that hidden embeddings encode an LLM's sense of "truthfulness", which may be steered along a vector of truth through test-time intervention. Self-reported confidence as explored by Manakul et al. (2023) and Kadavath et al. (2022) is a promising direction but requires the model to be well-calibrated and can suffer out-of-distribution.

## 6 CONCLUSION

Our study highlights the critical issue of hallucinations in Large Language Models (LLMs) and the importance of detecting these instances for safe deployment. We have established a link between hallucinations and model uncertainty, noting that existing methods primarily focus on next-token sampling as the sole source of randomness. Our investigation into the effects of injecting noise into the hidden states of intermediate layers reveals that introducing randomness at earlier stages of computation has a complementary impact on model uncertainty. By combining both intermediate layer randomness and prediction layer sampling, we propose an enhanced approach for hallucination detection. Extensive experiments validate the effectiveness of this combined scheme, demonstrating its potential to improve the reliability of LLMs.

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
