## A    ABLATION OF NOISE MAGNITUDE ON ALTERNATIVE DATASETS

In Section 4.1, we select $\alpha = 0.05$ for all datasets based on GSM8K performance and demonstrate that this choice remains effective across other datasets. However, the optimal noise magnitude varies by dataset. In Table 8, we ablate the noise magnitude for PrOntoQA and find that $\alpha = 0.08$ outperforms $\alpha = 0.05$. While our approach in Section 4.1 simplifies hyperparameter selection by choosing a single noise magnitude per model, these results suggest that tuning noise magnitude for each dataset individually could improve performance, albeit at the cost of additional computation.

Table 8: **Ablation of Noise Magnitude on ProntoQA.** Noise level $\alpha = 0.08$ further improves detection effectiveness compared to $\alpha = 0.05$, as indicated by a higher AUROC. Evaluation with `Llama2-13B-chat` model across 5 generations.

|  | Answer Entropy |
| --- | --- |
| noise magnitude = 0 | 65.07 |
| noise magnitude = 0.05 | 66.68 |
| noise magnitude = 0.08 | 67.58 |

## B    IMPLEMENTATION DETAILS

### B.1    DATASETS

We use in-context examples to demonstrate correct answer formatting and simplify answer extraction following free-form rationales, where applicable. For **GSM8K** and **CSQA**, we use the same prompts as in Wei et al. (2022). For **PrOntoQA**, Saparov & He (2023) generate a unique set of examples for each question. We extract the pre-generated prompts from the distributed model outputs. For **TriviaQA**, we ensemble a 10-shot prompt from the first 10-training examples of the format:

```
Q: Which Oscar-nominated film had You Sexy Thing as its theme
song?  A: The Full Monty Q: Which Joan's career revived in
Whatever Happened to Baby Jane?  A: Crawford Q: Which much-loved
actor won the Best Actor Oscar for The Philadelphia Story?  A:
James Stewart (...)  Q: In which river is the Boulder Dam?  A:
```

To address instances where the model maintains the Q:...A:... format after delivering an answer, we trim all generations using pattern matching with a set of stopwords identified in the outputs. In evaluation, when the model fails to produce the result in format, we consider the answer as invalid.

### B.2    MODELS

All models evaluated in this work are off-the-shelf with no additional fine-tuning. Perturbation on model is implemented using `pyvene` (Wu et al., 2024). We run all of our experiments on 80GB NVIDIA A100s. And there is no noticeable latency overhead with or without noise injection, confirming that our method introduces no practical delay.

## C    VISUALIZATION OF HALLUCINATION/NON-HALLUCINATION SEPARATION

In Figure 7, we visualize the enhancement of hallucination/non-hallucination separation with number of generations $K = 5$. In the following, we visualize the same for $K = 10, 15, 20$. Across all visualizations, we observe that injecting noise enhances the separation between hallucination and non-hallucination instances, improving the effectiveness of detection.

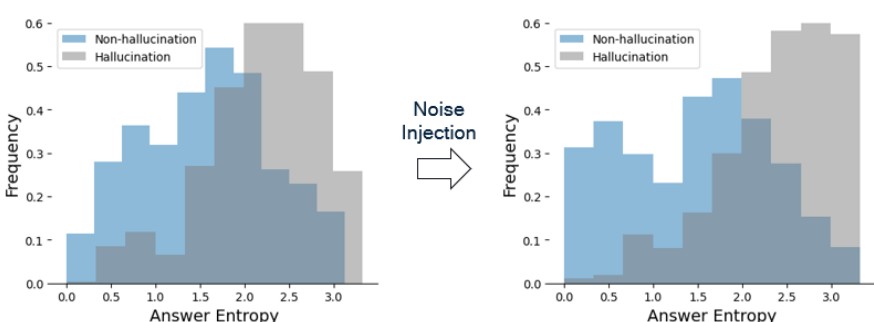

Figure 5: **Intermediate Layer Randomness Enhances Hallucination Detection.** Evaluation performed on GSM8K dataset with `Llama2-13B-chat` model across **10** generations. Rest of setup up follows Figure 7 (b)

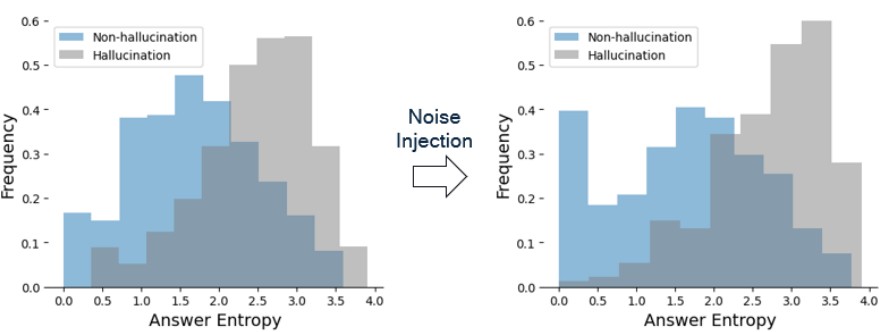

Figure 6: **Intermediate Layer Randomness Enhances Hallucination Detection.** Evaluation performed on GSM8K dataset with `Llama2-13B-chat` model across **15** generations. Rest of setup up follows Figure 7 (b)

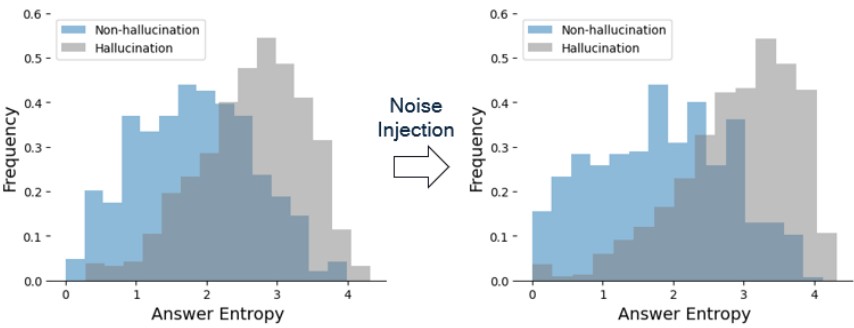

Figure 7: **Intermediate Layer Randomness Enhances Hallucination Detection.** Evaluation performed on GSM8K dataset with `Llama2-13B-chat` model across **20** generations. Rest of setup up follows Figure 7 (b)