# OpenReview forum: "Enhancing Hallucination Detection with Noise Injection"
_ICLR.cc/2025/Conference — Submitted to ICLR 2025_

### Official Review · Reviewer_76cD · 2024-10-19

**Soundness:** 2
**Presentation:** 3
**Contribution:** 2
**Rating:** 5
**Confidence:** 4

**Summary:**

This work builds upon the idea that the variability of LLM answers to a question is most pronounced when the LLM does not know the correct answer. By perturbing the intermediate LLM layers, they show this gap in variability tends to increase, facilitating the detection of hallucinations.

The work is largely empirical. Most of the results are shown for the GSM8K dataset, where the method appears to work best. On three other datasets, results are still positive but much more contained. Table 3 would benefit from reporting standard deviations over the multiple runs. Right now it is not clear if the difference in entropy over CSQA, TriviaQA and ProntoQA is significant.

I appreciate the insight this work brings in terms of showing that the epistemic uncertainty induced by perturbing intermediate layers can provide complementary effects to the aleatoric uncertainty induced by last layer for the purpose of detecting hallucinations. However, considering the complications introduced - the method needs access to the intermediate layers of the model, it may be sensitive to the noise magnitude (the Appendix in this direction is not particularly extensive) and to which layers are perturbed - I wonder if the improvements are in fact worth the effort.

I'd suggest the authors to provide a comprehensive evaluation across many datasets, including standard deviation of the results, to show that the method works robustly in multiple instances.

**Strengths:**

- Perturbing intermediate layers seems to increase the uncertainty gap between instances where the model is correct and where it is not.
- The authors make an effort in ablating their results, in particular to distinguish the noise effect induced by intermediate vs last layer.

**Weaknesses:**

- Results seem significant on GSM8K, less so on the other datasets. Standard deviations are missing.
- It may be worth extending the analysis on the sensitivity to the noise magnitude to better gauge the robustness of the algorithm. In the main paper, the authors only use either no noise or noise magnitudes 0.01 and 0.05, and only for one dataset. In the Appendix, results for another dataset are presented, but at different noise magnitudes. It would be good to provide results for a sufficient amount of noise magnitudes and all datasets.

**Questions:**

- The authors refer to Figure 7 multiple times throughout the text. I believe this is a type, as there is no Figure 7. Should this be Figure 2 instead?

---

> ### Author Response · Authors · 2024-11-26
>
> Thank you for your comments and suggestions. We apologize for the typo -- Figure 7 should be Figure 2 instead.  Please see our response below.
>
> **[Tradeoff of Complication and Performance]** Thank you for highlighting the trade-offs involved. We agree that the method introduces additional considerations. As such, the decision to adopt this approach would indeed depend on the specific application and its requirements. We appreciate your suggestion and acknowledge that future work could further explore these aspects to refine the effort-benefit tradeoff.
>
> Weaknesses
>
> **[Statistical Significance]** Thank you for raising this concern. To assess the statistical significance of our results, we report the 95% confidence intervals in the table below. Specifically, we use a bootstrap method to estimate the intervals: we sample five generations per question from a broader pool of 20 generations with replacement and a bootstrap sample size of 25 for GSM8K, TriviaQA, and CSQA. For ProntoQA, we use a bootstrap sample size of 50 due to higher data variability.
>
> | Metric                           | GSM8K                     | CSQA                      | TriviaQA                  | ProntoQA                 |
> |----------------------------------|:-------------------------:|:-------------------------:|:-------------------------:|:------------------------:|
> | Answer Entropy                   | 72.91 ± 0.43              | 68.70 ± 0.52              | 62.74 ± 0.10              | 65.45 ± 0.66             |
> | Answer Entropy w/ Noise          | 79.04 ± 0.44 (+6.13)      | 69.89 ± 0.47 (+1.19)      | 64.04 ± 0.10 (+1.30)      | 66.38 ± 0.64 (+0.93)     |
>
> We also conduct a t-test to determine whether the changes are statistically significant. All datasets pass the t-test (significance level: $\alpha = 0.05 $), with the following results: GSM8K ($ t_\text{crit} = 1.677, t_\text{score} = 20.669 $), CSQA ($ t_\text{crit} = 1.677, t_\text{score} = 3.553 $), TriviaQA ($( t_\text{crit} = 1.677, t_\text{score} = 19.259 $), and ProntoQA ($ t_\text{crit} = 1.661, t_\text{score} = 2.041 $).
>
> Given the relatively small answer space, calculating entropy using five samples provides sufficient precision to highlight the differences introduced by noise injection. For predictive probability and normalized predictive entropy—metrics that evaluate a significantly larger space encompassing all possible reasoning and answer sequences—five generations sampled from our pool of 20 are less likely to yield reliable Monte Carlo estimates. This limitation could potentially be addressed in future studies by increasing the number of samples, expanding the generation pool, or focusing exclusively on the answer string. Nonetheless, under our current setup, our experiments demonstrate that noise injection does not degrade performance under these measures.
>
> **[Sensitivity on Noise Magnitude]** Thank you for the suggestion. In response, we have conducted a sensitivity analysis of noise magnitude on both CSQA and TriviaQA datasets. While we observe that the optimal noise magnitude varies across datasets, the results indicate that noise injection over a broad range of magnitudes consistently improves performance. We will update the manuscript to include these experiments, providing results across multiple noise magnitudes to ensure a comprehensive evaluation.
> |     Noise   Magnitude    |     TriviaQA AUROC    |     CSQA   AUROC    |
> |:------------------------:|:---------------------:|:-------------------:|
> |             0            |          61.66        |         60.93       |
> |            0.01          |          62.06        |         61.70       |
> |            0.02          |          62.11        |         62.87       |
> |            0.03          |          62.29        |         63.34       |
> |            0.04          |          62.60        |         62.61       |
> |            0.05          |          63.18        |         62.69       |
> |            0.06          |          63.41        |         63.84       |
> |            0.07          |          63.96        |         62.99       |
> |            0.08          |          64.37        |         63.34       |
> |            0.09          |          64.83        |         63.48       |
> |            0.10          |          65.07        |         63.18       |
> Table: Sensitivity Analysis of Noise Magnitude on TriviaQA and CSQA.

---

> > ### Comment · Reviewer_76cD · 2024-12-02
> >
> > Thank you for addressing my concerns. Given also the comments from the other reviewers, I stand by my score. I believe the paper might benefit from trying to theoretically ground the approach. Furthermore, although the authors show statistically significant improvements on all datasets in the rebuttal, only on one dataset the benefits are very evident, and my question of whether the introduced additional complexity is worth remains.

---

> ### Author Response · Authors · 2024-12-02
>
> Thank you for your feedback. Regarding the theoretical grounding, we have addressed this concern by elaborating on how adjusting parameters like $T $, complements noise injection. Specifically, temperature modifies the sampling distribution while preserving the token likelihood order (e.g., $\text{Pr}(\text{token}_A) > \text{Pr}(\text{token}_B) $), whereas noise injection can reverse this order, offering a complementary effect. We have revised lines 80–84 and 226–229 in the manuscript to incorporate this discussion.
>
> As for the additional complexity, we  have already clarified that it represents a trade-off. Its value depends on the application context, and practitioners may weigh this against their specific goals and constraints. Moreover, the proposed approach does not introduce additional inference delay, ensuring practical applicability in latency-sensitive scenarios.

---

### Official Review · Reviewer_S4r6 · 2024-11-02

**Soundness:** 2
**Presentation:** 3
**Contribution:** 3
**Rating:** 5
**Confidence:** 4

**Summary:**

The paper proposes to inject noise in the intermediate representations to enhance hallucination detection. The method is mainly tested on Llama2 on 4 different datasets.

**Strengths:**

- The paper flows well with detailed explanations.
- Ablation experiments are thorough and extensive.
- The problem of Hallucination detection is crucial in recent LLM studies.

**Weaknesses:**

- My main concern is the soundness of the experimental results. Although the authors have shown the std of experiments in Figure 4, this is only shown for the dataset, GSM8K, which had the greatest improvement. However, considering that the gain in the other three datasets is relatively smaller, I would like to see the std values for other datasets too. Also, please conduct a t-test on the improvements.
- The authors tested their method mainly on Llama2-13B-chat. Although the experiment on Mistral has been provided in Table 6, this is only done on GSM8K. I would like to see a full table of experiments on other datasets.
- The message of Figure 2 (b) is somewhat unclear to me. I don't think the figures demonstrate better separability between non-hallucination and hallucination. Maybe a more fine-grained histogram would show a better picture?
- (minor) There are some grammatical issues in writing. I suggest using Grammarly or ChatGPT to refine the manuscript.
- (minor) There is no Figure 7 while the manuscript keeps referring to it. I'm assuming it should have been Figure 2, but please correct this.

Overall, the paper is well written. However, my main concern is the significance and generality of the approach. If my concerns are resolved, I would be happy to adjust my scores.

**Questions:**

See weaknesses.

---

> ### Author Response · Authors · 2024-11-26
>
> Thank you for your comments and suggestions. We apologize for the typo, and Figure 7 should be Figure 2 instead.  We cleaned up the typos in the updated draft. Please see our response below.
>
> Weaknesses
>
> **[statistical significance and t-test]** Thank you for raising this concern. To assess the statistical significance of our results, we report the 95% confidence intervals in the table below. Specifically, we use a bootstrap method to estimate the intervals: we sample five generations per question from a broader pool of 20 generations with replacement and a bootstrap sample size of 25 for GSM8K, TriviaQA, and CSQA. For ProntoQA, we use a bootstrap sample size of 50 due to higher data variability.
>
> | Metric                           | GSM8K                     | CSQA                      | TriviaQA                  | ProntoQA                 |
> |----------------------------------|:-------------------------:|:-------------------------:|:-------------------------:|:------------------------:|
> | Answer Entropy                   | 72.91 ± 0.43              | 68.70 ± 0.52              | 62.74 ± 0.10              | 65.45 ± 0.66             |
> | Answer Entropy w/ Noise          | 79.04 ± 0.44 (+6.13)      | 69.89 ± 0.47 (+1.19)      | 64.04 ± 0.10 (+1.30)      | 66.38 ± 0.64 (+0.93)     |
>
> We also conduct a t-test to determine whether the changes are statistically significant. All datasets pass the t-test (significance level: $\alpha = 0.05 $), with the following results: GSM8K ($ t_\text{crit} = 1.677, t_\text{score} = 20.669 $), CSQA ($ t_\text{crit} = 1.677, t_\text{score} = 3.553 $), TriviaQA ($( t_\text{crit} = 1.677, t_\text{score} = 19.259 $), and ProntoQA ($ t_\text{crit} = 1.661, t_\text{score} = 2.041 $).
>
> Given the relatively small answer space, calculating entropy using five samples provides sufficient precision to highlight the differences introduced by noise injection. For predictive probability and normalized predictive entropy—metrics that evaluate a significantly larger space encompassing all possible reasoning and answer sequences—five generations sampled from our pool of 20 are less likely to yield reliable Monte Carlo estimates. This limitation could potentially be addressed in future studies by increasing the number of samples, expanding the generation pool, or focusing exclusively on the answer string. Nonetheless, under our current setup, our experiments demonstrate that noise injection does not degrade performance under these measures.
>
> **[Mistral Experiments]** Thank you for the suggestion. In response to your request, we conducted additional TriviaQA experiments on Mistral, using noise magnitudes of 0 and 0.02, consistent with our GSM8K experiments on Mistral. Without noise injection, hallucination detection AUROC is 66.42, and with noise injection, AUROC improves to 69.44. Due to time and computational constraints, we focus on this representative evaluation, which further demonstrate the method’s generality.
>
> **[Figure Illustration]** Thank you for your suggestion. Figure 2 is based on only 5 generations, which limits the granularity of the entropy values. We are open to any further suggestions on enhancing visual understanding.

---

### Official Review · Reviewer_AUc5 · 2024-11-03

**Soundness:** 2
**Presentation:** 3
**Contribution:** 2
**Rating:** 5
**Confidence:** 4

**Summary:**

This paper proposes enhancing the performance of hallucination detection by perturbing hidden unit activations in intermediate layers for sampling-based methods. Unlike existing approaches that measure uncertainty through prediction layer sampling, this work introduces noise to intermediate layer representations and combines this noise injection with prediction layer sampling to improve hallucination detection. Extensive experiments demonstrate the effectiveness of this method across various datasets, uncertainty metrics, and model architectures.

**Strengths:**

1. The motivation for introducing randomness in the hidden layer is intuitive and makes a lot of sense. The paper is well-written and easy to implement.
2.  The concept of perturbing intermediate representations to enhance the separability between hallucinated and non-hallucinated generation is overall innovative.
3. Extensive experiments are provided to demonstrate the effectiveness of noise injection in enhancinghallucination detection across various datasets and uncertainty metrics.

**Weaknesses:**

1. The performance improvement from noise injection is insignificant in most cases. As illustrated in Table 3, there is an insignificant increase in Predictive Entropy and Normalized Entropy, with the most notable improvement occurring only in the answer entropy of the GSM8K dataset.
2. The author argues that the effects of noise injection and prediction layer sampling are complementary.  However, this claim is not strongly substantiated by the results shown in Figure 3. A Pearson correlation of 0.67 does not clearly indicate a complementary relationship between these two sources of randomness. Even without introducing noise, drawing entropy with temperatures T=0.5 and T=1.0 will show similar positive correlations.
3. The author introduced additional hyperparameters $\alpha$, $\ell_1$ and $\ell_2$  to adjust the randomness of sampling. However, this comparison may be unfair, as performance could also be enhanced by optimizing parameters such as temperature T, top_P, and top_K.
4. Theoretical insight is limited in explaining why perturbations at the hidden layer are more effective than output layer sampling for self-consistency based hallucination detection methods. In my opinion, using a larger temperature is essentially the same as modifying the feature space to increase randomness.

**Questions:**

1. Is there any explanation why the performance is more significant only when combined with Answer Entropy?
2. I like the results shown in Table 4, but I would appreciate it if the author can proivde more experiments in other datasets, such as CSQA or TriviaQA.
3. I would like to see more perturbation based methods. For example, what will happen if we perturb the input query for those samping based methods?

---

> ### Author Response · Authors · 2024-11-26
>
> Thank you for your valuable questions and comments! Please see our response below.
>
> Weaknesses
>
> **[Statistical Significance]** Thank you for raising this concern. To assess the statistical significance of our results, we report the 95% confidence intervals in the table below. Specifically, we use a bootstrap method to estimate the intervals: we sample five generations per question from a broader pool of 20 generations with replacement and a bootstrap sample size of 25 for GSM8K, TriviaQA, and CSQA. For ProntoQA, we use a bootstrap sample size of 50 due to higher data variability.
>
> | Metric                           | GSM8K                     | CSQA                      | TriviaQA                  | ProntoQA                 |
> |----------------------------------|:-------------------------:|:-------------------------:|:-------------------------:|:------------------------:|
> | Answer Entropy                   | 72.91 ± 0.43              | 68.70 ± 0.52              | 62.74 ± 0.10              | 65.45 ± 0.66             |
> | Answer Entropy w/ Noise          | 79.04 ± 0.44 (+6.13)      | 69.89 ± 0.47 (+1.19)      | 64.04 ± 0.10 (+1.30)      | 66.38 ± 0.64 (+0.93)     |
>
> We also conduct a t-test to determine whether the changes are statistically significant. All datasets pass the t-test (significance level: $\alpha = 0.05 $), with the following results: GSM8K ($ t_\text{crit} = 1.677, t_\text{score} = 20.669 $), CSQA ($ t_\text{crit} = 1.677, t_\text{score} = 3.553 $), TriviaQA ($( t_\text{crit} = 1.677, t_\text{score} = 19.259 $), and ProntoQA ($ t_\text{crit} = 1.661, t_\text{score} = 2.041 $).
>
> Given the relatively small answer space, calculating entropy using five samples provides sufficient precision to highlight the differences introduced by noise injection. For predictive probability and normalized predictive entropy—metrics that evaluate a significantly larger space encompassing all possible reasoning and answer sequences—five generations sampled from our pool of 20 are less likely to yield reliable Monte Carlo estimates. This limitation could potentially be addressed in future studies by increasing the number of samples, expanding the generation pool, or focusing exclusively on the answer string. Nonetheless, under our current setup, our experiments demonstrate that noise injection does not degrade performance under these measures.
>
> **[Complementary Effect and Pearson Correlation]**  We agree that sampling at different temperatures may yield a similar Pearson correlation. However, such correlations still reflect complementary effects between different sampling temperatures. While combining different temperatures to leverage their complementary effect is not straightforward, our algorithm demonstrates how noise injection and temperature-based sampling can be effectively combined to leverage their complementary effects.
>
> **[Fair Comparison against Adjusting T, Top-P, Top-K]**  We agree that performance could be enhanced by optimizing hyperparameters such as $T $, $\text{top-P} $, and $ \text{top-K} $. To ensure a fair comparison, we present results with different $T $ values while using the default values for$ \text{top-P}$ and $ \text{top-K} $ in Table 4. We observe that noise injection improves performance across all tested temperatures.
> Exhaustively testing all possible configurations is infeasible. However, theoretically, adjusting $ T $, $ \text{top-P} $, and $ \text{top-K} $ would complement noise injection. Specifically, these parameters alter the sample distribution but preserve token likelihood ordering (e.g., if $ \text{Pr}(token_A) > \text{Pr}(token_B) $, this order remains unchanged). In contrast, noise injection can reverse this order, offering a complementary effect.
>
> **[Theoretical Insight and Comparison with Increasing T]** Thank you for the feedback. We do not claim that perturbations at the hidden layer are more effective than output layer sampling. Instead, we suggest that combining both can be more effective due to their complementary effects, as discussed theoretically in our response to [Adjusting T, Top-P, Top-K] . We also edit line 80-84, 226-229 to include the discussion.
> Empirically, we show in Table 4 that noise injection and temperature adjustments have distinct effects. Specifically, increasing \( T \) from 0.5 to 0.8 or 1.0 reduces hallucination detection AUROC, but adding noise at \( T = 0.5 \) introduces a different source of randomness and improves performance (lines 425–428).

---

> ### Author Response · Authors · 2024-11-26
>
> Questions:
>
> **[Explanation of Better Significance on Answer Entropy]** We address the question in response to [Statistical Significance]
>
> **[Ablation study in Table 4 on additional datasets]** Thank you for your suggestions. Below we ablate on temperature and noise magnitude with CSQA dataset. As in Table 4 in the paper, noise injection improves detection effectiveness compared to no noise. We will updated the manuscript to include the results.
> |                |     Noise Magnitude = 0    |     Noise Magnitude =   0.01    |     Noise Magnitude= 0.05    |
> |----------------|----------------------------|---------------------------------|------------------------------|
> |     T = 0.2    |     60.93                  |     61.70                       |     62.69                    |
> |     T = 0.5    |     65.49                  |     67.56                       |     67.38                    |
> |     T = 0.8    |     68.70                  |     68.94                       |     69.89                    |
> |     T = 1.0    |     69.42                  |     71.58                       |     70.14                    |
> Table: Ablation on Temperature and Noise Magnitude. Evaluation on CSQA dataset with Llama2-13B-chat model across 5 generations.
>
> **[Other Perturbation Methods]** Thank you for the suggestion. While input query perturbations are interesting, they are beyond the scope of this work. We focus on model perturbations and leave this direction for future exploration.

---

### Official Review · Reviewer_drSq · 2024-11-03

**Soundness:** 2
**Presentation:** 2
**Contribution:** 2
**Rating:** 3
**Confidence:** 5

**Summary:**

The paper addresses the challenge of detecting "hallucinations" in Large Language Models (LLMs). The study proposes a novel technique to improve hallucination detection by adding "noise injection" to intermediate layers of the model, creating an additional source of randomness during response generation.

**Strengths:**

- The paper touches a critical issue in current LLMs. Any progress in error detection is critical to the field.

**Weaknesses:**

The paper presents some notable weaknesses in both the presentation of content and in aspects of the methodology and experimental design. Below are specific areas of concern:

- The review of related work is somewhat shallow. There is substantial literature on detecting hallucinations in models, yet this paper does not adequately differentiate its approach or clarify how it builds upon existing insights.
- All experiments are conducted on a single model, which limits the generalizability of the conclusions. Testing across multiple models would strengthen the claims.

## Intro:
- The term "hallucinations" is only briefly defined as instances where a model generates “plausible yet incorrect responses.” However, it remains unclear if this term includes all model errors or just those based on plausibility. The paper does talk about plausibility further, leaving the reader uncertain about what qualifies as a hallucination.
- You refer to figure 7  which is in the appendix. Core results should be presented in the main paper, and anything you talk about in the intro is definitely core. Note that reviewers are not required to read them but in your case it was fundamental to understand your results. This note is relevant for the rest of the paper as well.
- We empirically validate the hypothesis in Figure 7 (a) -> how exactly the figure validates your hypothesis? Readers need a step-by-step walkthrough to see how Figure 7(a) substantiates the hypothesis.

## Section 2:

- The definition of $f$ is a bit vague and as a results, the method as well. The model's output is not a function of all of its hidden states, because each hidden state $l$ is a function of the previous hidden state $l-1$. I think that maybe you could say that if you talk about the residual stream that sums all hidden states (because later you talk about mlp output), but it is very not clear at this point of reading.
- Because of that, it's not clear what happens when you replace $h_t^l$ with a noised version. Do you recompute $h_t^{l+1}$ to get a noised version or do you just noise the clean version? This needs to be clearly explained. If you add the noise to the MLP output which in turn simply goes to the residual stream, and you don't recompute the following MLPs in higher layers after adding noise, then this is just equivalent to add noise K times (where K are the number of layers you noised) to the residual stream, without significance the the specific layers that are noised, because the unembedding layer simply takes the residual stream after the final layer.

## Section 3:

- Table 2 lacks information on statistical significance, including standard deviations and the number of seeds used for experiments. Additionally, there is no indication of the dataset size.
- he statement, “This supports our intuition that incorrect answers are less robust to noise injection…” appears without prior context. While there is mention of hallucinations having higher entropy, there is no discussion that wrong answers may appear less after noise injections. Why does this happens?
- It was not clear to me why you need a separate section for GSM8K as experiments are later conducted across multiple datasets, making this section feel repetitive.

## Section 4:

The paper lacks a clear presentation of noise boundaries and statistical significance tests, which raises concerns about the reliability of findings. The difference between the proposed methods and baselines is small, and it is unclear how significant these differences are. Only Figure 4 provides such comparisons for GSM8K, while other datasets are not covered.

Some other typos etc.:
- Links to figures/equations are broken.
- Line 118: "**an** uncertainty metric"
- Line 122 sentence is not grammatically correct
- Line 289 ".,"
- Figure 7 caption: "Rest of setup up follows Figure 7 (b)" -> typo?


I believe that all of these issue could be fixed in a revision (not sure that in the short time of the rebuttal period) and then it will be a valuable research paper.

**Questions:**

- How do you extract the final answers from the long answer? How do you make sure it is always in the end? Do you do some sort of prompt engineering or few shot for this
- What is the acc of the model in greedy decoding?
- Why are the results on GSM8K are different in table 2 and 3? What is the difference in the setting?
- "For each dataset, we select the temperature within T = {0.2, 0.5, 0.8, 1.0} which optimizes the model accuracy on this dataset" - on the validation dataset?

---

> ### Author Response · Authors · 2024-11-26
>
> Thank you for your comprehensive review and constructive comments. We apologize for the typos and Figure 7 should instead be Figure 2. We have fixed the typos in the updated PDF. Please see our response below.
>
> Weaknesses
>
> **[Related work]** Thank you for your suggestion. Our work builds upon prior studies linking model uncertainty to hallucination (lines 31-35; lines 495-500) by introducing a new source of randomness. Since we modify intermediate layers, we also review relevant work on hallucination detection using intermediate layer representations (lines 501-508). While we focus on areas most relevant to our contributions, we welcome any specific references to further enrich our discussion.
>
> **[Single Model]**  Kindly note that we experimented with multiple models:  LLAMA2-13B-Chat for our case study and alternative models -- LLAMA2-13B-Chat and Mistral -- in Table 6.
>
> **[Intro: Hallucination Definition]** Thank you for raising this concern.  In our experimental setup, model responses are coherent (i.e., not repetitive or unreadable) and their correctness cannot be determined without reference (see Line 3 in Table 1). In this context, incorrect answers are considered plausible, which aligns with our definition of hallucinations. We have updated line 161-170 to clarify this.
>
> **[Intro: Empirical Validation]** Thank you for raising this concern. We empirically validate our hypothesis that hallucination cases are less robust to noise injection in Figure 2(a). Specifically, the hallucination cases (grey) exhibit higher entropy, indicating greater variance in responses with injected noise. We have updated line 89-90 to clarify this point.
>
> **[Section 2: Hidden State Perturbation]**  Thank you for raising this concern. To clarify, when h_l is perturbed, we first compute h_{l+1} from the perturbed h_l. If layer l + 1 is selected, we then apply noise to h_{l+1}. Thus, the noise is not only added to the residual stream. We have updated line 143-144, 268-269, 298-299 to better explain this process.
>
> **[Section 3: Significance and dataset size]** Table 2, as a case study example, is over one single seed. For the same setup, we demonstrate the standard deviation with 20 random seeds Figure 4.  We evaluate on GSM8K test set containing 1319 questions. We have updated line 160-161 to report the dataset size.
>
> **[Section 3: Noise Effect on Accuracy.]**  Thank you for raising this concern. Kindly note that we do not claim that **within a single generation**, the number of hallucination cases appear less with noise injection. Instead, we argue that the incorrect answers generated during hallucination are **less consistent across generations** with noise injected, making incorrect answers less likely to be selected by majority vote. As a result, this shift improves the likelihood of correct answers being chosen, thereby enhancing accuracy under the majority vote scheme. We update line 338-344 in the manuscript to clarify the explanation.
>
> **[Section 3: Separate section for GSM8K]** Thank you for your feedback. Section 3 serves as our methodology section, detailing our approach through a case study on GSM8K. Section 4, by contrast, presents experimental results across multiple datasets. This distinction aims to improve clarity, but we are open to suggestions for reducing potential repetition.

---

> ### Author Response · Authors · 2024-11-26
>
> **[Section 4: Statistical Significance]**  Thank you for raising this concern. To assess the statistical significance of our results, we report the 95% confidence intervals in the table below. Specifically, we use a bootstrap method to estimate the intervals: we sample five generations per question from a broader pool of 20 generations with replacement and a bootstrap sample size of 25 for GSM8K, TriviaQA, and CSQA. For ProntoQA, we use a bootstrap sample size of 50 due to higher data variability.
>
> | Metric                           | GSM8K                     | CSQA                      | TriviaQA                  | ProntoQA                 |
> |----------------------------------|:-------------------------:|:-------------------------:|:-------------------------:|:------------------------:|
> | Answer Entropy                   | 72.91 ± 0.43              | 68.70 ± 0.52              | 62.74 ± 0.10              | 65.45 ± 0.66             |
> | Answer Entropy w/ Noise          | 79.04 ± 0.44 (+6.13)      | 69.89 ± 0.47 (+1.19)      | 64.04 ± 0.10 (+1.30)      | 66.38 ± 0.64 (+0.93)     |
>
> We also conduct a t-test to determine whether the changes are statistically significant. All datasets pass the t-test (significance level: $\alpha = 0.05 $), with the following results: GSM8K ($ t_\text{crit} = 1.677, t_\text{score} = 20.669 $), CSQA ($ t_\text{crit} = 1.677, t_\text{score} = 3.553 $), TriviaQA ($( t_\text{crit} = 1.677, t_\text{score} = 19.259 $), and ProntoQA ($ t_\text{crit} = 1.661, t_\text{score} = 2.041 $).
>
> Given the relatively small answer space, calculating entropy using five samples provides sufficient precision to highlight the differences introduced by noise injection. For predictive probability and normalized predictive entropy—metrics that evaluate a significantly larger space encompassing all possible reasoning and answer sequences—five generations sampled from our pool of 20 are less likely to yield reliable Monte Carlo estimates. This limitation could potentially be addressed in future studies by increasing the number of samples, expanding the generation pool, or focusing exclusively on the answer string. Nonetheless, under our current setup, our experiments demonstrate that noise injection does not degrade performance under these measures.
>
> Questions:
>
> **[Prompting and Answer Extraction]** We prompt the model using In-Context Learning examples and extract the final answers based on the formatting provided in these examples. We provide further details in Appendix B.1.
>
> **[Greedy Decoding Accuracy]**  With greedy decoding GSM8K reports 29.11% accuracy; CSQA reports 62.62% accuracy; TriviaQA reports 73.57% accuracy; ProtoQA reports 76.2% accuracy.
>
> **[Difference on Table 2 and Table 3]** The results differ because Table 2 presents GSM8K performance under a single random seed, which is the same seed used in Figures 2 and 3 for the corresponding experiments. In contrast, Table 3 reports the average performance across 20 random seeds.

---

### Official Review · Reviewer_8zQH · 2024-11-11

**Soundness:** 2
**Presentation:** 2
**Contribution:** 2
**Rating:** 5
**Confidence:** 4

**Summary:**

This paper explores the potential of injecting noise into the intermediate layer outputs of LLMs to induce greater uncertainty when they are prone to hallucination.

**Strengths:**

* Good logical flow and storytelling.
* Clear presentation of experimental results and straightforward mathematical formulations.

**Weaknesses:**

* Lack of theoretical justification for the noise injection approach: Although the injection method is simplistic, the authors do not clarify why they chose to sample noise from a uniform distribution with fixed mean and variance across LLMs. This choice raises concerns about the generalizability of the results.
* No evaluation of statistical significance: The reported performance improvements with noise injection are marginal, and the absence of confidence intervals weakens claims regarding these improvements.

Overall, I feel that this paper is still not ready for publication.

**Questions:**

No specific question from me. But my concerns are majorly stated in the previous section.

---

> ### Author Response · Authors · 2024-11-26
>
> Thank you for your comments and suggestions. Please see our response to your concerns below.
>
> **[Fixed Distribution]** While we exemplify our method with a uniform distribution, the mean and variance **do** vary as we select different noise magnitudes. As clarified in lines 464–466 of Section 4.5, the sampling distribution depends on the specific LLM, enabling adaptation and supporting generalizability.
>
> **[Statistical Significance]** Thank you for raising this concern. To assess the statistical significance of our results, we report the 95% confidence intervals in the table below. Specifically, we use a bootstrap method to estimate the intervals: we sample five generations per question from a broader pool of 20 generations with replacement and a bootstrap sample size of 25 for GSM8K, TriviaQA, and CSQA. For ProntoQA, we use a bootstrap sample size of 50 due to higher data variability.
>
> | Metric                           | GSM8K                     | CSQA                      | TriviaQA                  | ProntoQA                 |
> |----------------------------------|:-------------------------:|:-------------------------:|:-------------------------:|:------------------------:|
> | Answer Entropy                   | 72.91 ± 0.43              | 68.70 ± 0.52              | 62.74 ± 0.10              | 65.45 ± 0.66             |
> | Answer Entropy w/ Noise          | 79.04 ± 0.44 (+6.13)      | 69.89 ± 0.47 (+1.19)      | 64.04 ± 0.10 (+1.30)      | 66.38 ± 0.64 (+0.93)     |
>
> We also conduct a t-test to determine whether the changes are statistically significant. All datasets pass the t-test (significance level: $ \alpha = 0.05 $), with the following results: GSM8K ($ t_\text{crit} = 1.677, t_\text{score} = 20.669$), CSQA ($ t_\text{crit} = 1.677, t_\text{score} = 3.553 $), TriviaQA ($ t_\text{crit} = 1.677, t_\text{score} = 19.259 $), and ProntoQA ($ t_\text{crit} = 1.661, t_\text{score} = 2.041 $).
>
> Given the relatively small answer space, calculating entropy using five samples provides sufficient precision to highlight the differences introduced by noise injection. For predictive probability and normalized predictive entropy—metrics that evaluate a significantly larger space encompassing all possible reasoning and answer sequences—five generations sampled from our pool of 20 are less likely to yield reliable Monte Carlo estimates. This limitation could potentially be addressed in future studies by increasing the number of samples, expanding the generation pool, or focusing exclusively on the answer string. Nonetheless, under our current setup, our experiments demonstrate that noise injection does not degrade performance under these measures.

---

> > ### Comment · Reviewer_8zQH · 2024-12-03
> >
> > I appreciate the authors' responses!
> >
> > For the first issue, I would really like to see more formalized methodologies at adjusting variances; the current optimality results seemed not convincing as they are highly dependent on the variance chosen, and the choices seemed pretty random and I doubt its generalization.
> >
> > For the second issue, I am still concerned about the sample sizes being only 5 for computing the answer entropy. I hope the authors could increase the sample size like promised in the next iteration of this paper. Looking forward to the improvements!
> >
> > For the issues are not full addressed, I would keep my original score.

---

> > > ### Author Response · Authors · 2024-12-03
> > >
> > > Thank you for your thoughtful feedback!
> > >
> > > For the first concern, we appreciate your suggestion. Our current approach selects the per-model variance magnitude based on GSM8K and applies it consistently across other datasets. We will reinforce this point in the manuscript to ensure clarity. While we find this approach effective, we are open to further suggestions for refining the variance selection. Additionally, our results do not heavily depend on the specific variance chosen. For your reference, the sensitivity analysis table below demonstrates the robustness of our findings across a range of variance values. While we observe that the optimal noise magnitude varies across datasets, the results indicate that noise injection over a broad range of magnitudes consistently improves performance.
> > >
> > > |     Noise   Magnitude    |     TriviaQA AUROC    |     CSQA   AUROC    |
> > > |:------------------------:|:---------------------:|:-------------------:|
> > > |             0            |          61.66        |         60.93       |
> > > |            0.01          |          62.06        |         61.70       |
> > > |            0.02          |          62.11        |         62.87       |
> > > |            0.03          |          62.29        |         63.34       |
> > > |            0.04          |          62.60        |         62.61       |
> > > |            0.05          |          63.18        |         62.69       |
> > > |            0.06          |          63.41        |         63.84       |
> > > |            0.07          |          63.96        |         62.99       |
> > > |            0.08          |          64.37        |         63.34       |
> > > |            0.09          |          64.83        |         63.48       |
> > > |            0.10          |          65.07        |         63.18       |
> > > Table: Sensitivity Analysis of Noise Magnitude on TriviaQA and CSQA.
> > >
> > > For the second concern, we note that a sample size of 5 falls within the range used in well-established prior work (see Figure 3(a) in [1]). In addition, in Figure 4 in our manuscript, we have already explored higher sample sizes and found the results to be consistent. While further increasing the number of samples could offer additional insights, it also introduces significant computational cost, which is an important practical consideration.
> > >
> > > We hope this clarifies our choices and demonstrates the validity of our approach.
> > >
> > > [1] Kuhn, Lorenz, Yarin Gal, and Sebastian Farquhar. "Semantic uncertainty: Linguistic invariances for uncertainty estimation in natural language generation." arXiv preprint arXiv:2302.09664 (2023).

---

> > > > ### Comment · Reviewer_8zQH · 2024-12-03
> > > >
> > > > Thank you for your prompt responses.
> > > >
> > > > For the first issue, may I have an explanation regarding why with the magnitude of noise increasing, the performances on TriviaQA consistently increases, while for CSQA, the effects are more nuanced and probably less straightforward to make sense of?

---

> > > > > ### Author Response · Authors · 2024-12-03
> > > > >
> > > > > Thank you for your follow-up question. Below is our current understanding based on the task characteristics.
> > > > >
> > > > > For CSQA, as a multiple-choice task, responses must adhere to a specific format to be valid. Introducing randomness, either through sampling temperature or noise injection, can sometimes produce invalid outputs, leading to irregularities in the performance trend.
> > > > >
> > > > > In contrast, TriviaQA is a free-form question-answering task where all responses are treated as valid. This could explain the smoother performance improvements observed with increasing noise magnitude.
> > > > >
> > > > > While the optimal noise magnitude varies across datasets, as previously noted, noise injection consistently enhances performance across tasks. We appreciate your insightful question and hope this helps provide further understanding into the versatility of the tasks and the observed trends.

---

> > > > > > ### Comment · Reviewer_8zQH · 2024-12-03
> > > > > >
> > > > > > Thank you for your further explanation!
> > > > > >
> > > > > > I would like to increase my score to 5, but unfortunately no higher.

---

> > > > > > > ### Author Response · Authors · 2024-12-03
> > > > > > >
> > > > > > > Thank you for your thoughtful engagement and for revisiting your score. We truly appreciate the time and effort you've put into reviewing our work and providing valuable feedback. If there are any additional suggestions or areas you feel we could further refine, we'd be glad to take them into account for future iterations.

---

### Comment · Area_Chair_YeEB · 2024-11-26

Dear Reviewers,

The authors have responded - please see if they have addressed your concerns and engage in further discussion to clarify any remaining issues.

Thanks!
AC

---

> ### Comment · Area_Chair_YeEB · 2024-12-01
>
> Dear Reviewers,
>
> The authors have provided responses - do have a look and engage with them in a discussion to clarify any remaining issues as the discussion period is coming to a close in less than a day (2nd Dec AoE for reviewer responses).
>
> Thanks for your service to ICLR 2025.
>
> Best,
> AC

---

### Meta-Review · Area_Chair_YeEB · 2024-12-21

**Metareview:**

This paper proposes a new method for detecting when an LLM is hallucinating based on the empirical observation that higher model uncertainty is correlated with hallucinations. Specifically, the paper proposes introducing noise into intermediate layers of the model to amplify the separation between hallucinated and non-hallucinated outputs on uncertainty measures. While the approach is new and interesting, and the paper includes relevant ablation results, the empirical evaluation though promising, is insufficient to convincingly demonstrate the generalizability of the approach. The method was only evaluated on a limited combination of model architectures and datasets (initially 1 architecture x 4 datasets and separately 2 architectures x 1 dataset), limited improvements were seen in several cases, and no theoretical justifications are provided. Thus, overall the paper falls below the bar of acceptance for ICLR, though the authors are encouraged to strengthen their empirical evaluations and resubmit to a future conference.

**Additional Comments On Reviewer Discussion:**

Reviewers all appreciated the importance of the hallucination detection problem being solved and the promising results but all had concerns about the empirical evaluations, in terms of whether the improvements observed were significant and whether they generalize to other model architectures and tasks. The authors provided additional results including standard deviations and statistical tests during the rebuttal and one additional evaluation with Mistral, but these were not sufficient to convince the reviewers as there was also no theoretical justification provided (either in the original paper or the revision). Some other issues with presentation (drSq) and further analysis on the noise used (76cD, 8zQH) were addressed by the rebuttal. Overall, the lack of a comprehensive evaluation on multiple datasets and model combinations to convincingly demonstrate the generalizability of the method is a major limitation of the current work and needs to be addressed before the paper is ready for publication.

---

### Decision · Program_Chairs · 2025-01-22

Reject